# Exploration of Functional Connectivity Changes Previously Reported in Fibromyalgia and Their Relation to Psychological Distress and Pain Measures

**DOI:** 10.3390/jcm9113560

**Published:** 2020-11-05

**Authors:** Helene van Ettinger-Veenstra, Rebecca Boehme, Bijar Ghafouri, Håkan Olausson, Rikard K. Wicksell, Björn Gerdle

**Affiliations:** 1Department of Clinical Neuroscience, Karolinska Institute, S-171 77 Stockholm, Sweden; rikard.wicksell@ki.se; 2Center for Medical Image Science and Visualization (CMIV), Linköping University, S-581 85 Linköping, Sweden; hakan.olausson@liu.se (H.O.); bjorn.gerdle@liu.se (B.G.); 3Center for Social and Affective Neuroscience (CSAN), Linköping University, S-581 85 Linköping, Sweden; 4Pain and Rehabilitation Centre, and Department of Health, Medicine and Caring Sciences, Linköping University, S-581 85 Linköping, Sweden; bijar.ghafouri@liu.se

**Keywords:** insula, central executive network, intraparietal sulcus, resting state, anxiety, pain intensity, fMRI

## Abstract

Neural functional connectivity changes in the default mode network (DMN), Central executive network (CEN), and insula have been implicated in fibromyalgia (FM) but stem from a sparse set of small-scale studies with limited power for the investigation of confounding effects. We investigated whether anxiety, depression, pain sensitivity, and pain intensity modulated functional connectivity related to DMN nodes, CEN nodes, and insula. Resting-state functional magnetic resonance imaging data were collected from 31 females with FM and 28 age-matched healthy controls. Connectivity was analysed with a region-based connectivity analysis between DMN nodes in ventromedial prefrontal cortex (vmPFC) and posterior cingulate cortex, CEN nodes in the intraparietal sulcus (IPS), and bilateral insula. FM patients displayed significantly higher levels of anxiety and depressive symptoms than controls. The right IPS node of the CEN showed a higher level of connectivity strength with right insula in FM with higher pain intensity compared to controls. More anxiety symptoms in FM correlated with higher levels of connectivity strength between the vmPFC DMN node and right sensorimotor cortex. These findings support the theory of altered insular connectivity in FM and also suggest altered IPS connectivity in FM. Interestingly, no change in insular connectivity with DMN was observed.

## 1. Introduction

Fibromyalgia (FM) is defined as chronic (>3 months) pain widespread over the body with generalised hyperalgesia for mechanical pressure [1,2,3]. FM patients often experience depressive or anxiety symptoms [4]. As of yet, the underlying driving mechanisms of this disabling disorder remain largely unknown [5] and investigations of associated altered functional connectivity in the central nervous system are scarce and inconclusive [6]. Moreover, pain and psychological symptoms have shown to affect functional connectivity [6,7,8,9,10] and may thus confound functional connectivity measures in FM.

Functional connectivity changes can be investigated in intrinsically active neural networks consisting of interconnected regions; activation in these networks can be observed during resting state functional magnetic resonance imaging (rs-fMRI). Most commonly, a model-free analysis of within-network connectivity with independent component analysis (ICA), or a-priori region-based models of expected connectivity of anatomic or functional seeds can be applied. Two intrinsic neural networks thought to be affected by chronic pain conditions including FM are the self-referential default mode network (DMN) and the fronto-parietal central executive network (CEN) [11,12,13]. Also, the insula has shown involvement in different types of pain processing [14] and insular connectivity may be altered in FM [15]. In healthy controls (HC), the posterior cingulate cortex node of the DMN has a strong positive connection with the ventromedial prefrontal cortex (vmPFC) node of the DMN and a strong negative correlation with the insula and the intraparietal sulcus (IPS, a node of the CEN) [11]. In a study on altered functional connectivity in FM (*n* = 18), the DMN as a whole showed a more positive connection with the left anterior and middle insula and somatosensory cortex for FM when compared to HC (*n* = 18) [16]. Also, higher levels of connectivity strength within the right CEN component in the IPS was observed in FM compared to HC [16]. Another study showed that FM (*n* = 18) had less connectivity than HC (*n* = 18) between the left anterior insula and a cluster of voxels in the bilateral medial prefrontal gyrus, which partly overlapped with the vmPFC node of the DMN [17]. From these scarce results, there is no clear hypothesis on whether the DMN and CEN networks as a whole are affected in FM or that only specific network nodes show affected functional connectivity.

As a complicating factor for functional connectivity investigations in FM, several studies have shown that functional connectivity between the DMN and the CEN, the vmPFC, and insula can be affected by pain intensity; an increase in functional connectivity as an effect of experimental pain was found of right anterior insula with the anterior cingulate cortex in FM (*n* = 12) compared to HC (*n* = 15) [18]. Greater spontaneous pain intensity during an rs-fMRI scanning session was correlated to a higher level of connectivity strength of right anterior and middle insula with the DMN, and a higher level of connectivity strength of both the left and right insula with the CEN [16]. Diminished pain intensity after therapy in FM (*n* = 17) was associated with reduced connectivity between the DMN and the right insula [19]. Higher pain intensity in three different patient groups with either chronic low back pain (*n* = 18), complex regional pain syndrome (*n* = 19) or knee osteoarthritis (*n* = 14) was associated with a decreased connectivity of the vmPFC with the posterior components of the DMN [20]. These patients also showed an increased connectivity of the vmPFC with the insula [20].

Functional connectivity can also be affected by psychological distress, like depression and anxiety, which have shown to alter within-network DMN connectivity and connectivity of the whole DMN, of DMN and CEN nodes, and of insula with each other and with other regions [9,21,22]. In FM, depression interacts with chronic pain [23]; e.g., depression is associated with functional connectivity changes as an effect of experimental pain [18]. However, these associations have not been observed consistently [4]. 

The current study aimed to investigate a relatively large group of FM patients using rs-fMRI on functional connectivity changes associated with nodes in the DMN, the CEN and the insula. The chosen seed nodes for a region-based analysis were based on repeated findings in literature [16,17,18,20] and included: the vmPFC and posterior cingulate cortex nodes of the DMN, and the bilateral IPS nodes of the CEN. First, reproducibility of these previous findings of altered DMN node, CEN node, and insula connectivity was explored in interaction with measures of psychological distress, pain sensitivity and pain intensity. Second, an exploration of altered connectivity of DMN-nodes, CEN-nodes or insula with the whole brain was performed. 

## 2. Experimental Section

### 2.1. Subjects

Thirty-three female FM patients and 31 age-matched female HC between 22 and 56 years of age were recruited for a large-scale investigation on FM. The HC group was recruited through advertisements in newspapers, and the FM group consisted of patients from the Pain and Rehabilitation Centre at the University Hospital in Linköping, Sweden. Only females were included due to their overrepresentation in the patient base. Imaging data from two FM and three HC were missing or of too low quality, therefore, the final sample consisted of 31 FM and 28 HC participants, (age mean = 40.9 ± 10.9 standard deviation, range = 22–56 years). The study was granted ethical clearances by Linköping University Ethics Committee (Dnr: 2016/239-31); all participants gave their written informed consent, and the study was performed in accordance with the Helsinki Declaration.

### 2.2. Procedures

At the first visit, all participants underwent a clinical examination including pain sensitivity testing. After the first visit, all participants answered a set of health questionnaires covering demographic data as well as pain intensity, spreading and duration and psychological characteristics at home. During a second visit, blood and saliva samples were taken and a microdialysis was performed; this is not analysed in the current study. At the third visit, fMRI and spectroscopy data were obtained, including an rs-fMRI session of 10 min used in the current study. During a fourth visit, the nociceptive withdrawal reflex was tested; this is not analysed in the current study. The first and third visit were on average 3.8 months apart (SD = 2.98), this was an effect of scheduling constraints for the MR-scanner (Siemens Healthcare, Erlangen, Germany) and other equipment involved in other parts of this study as well as for the participants. No experimental treatment was given during this large-scale study, the patients continued their standard therapy. Exclusion criteria included an inability to refrain from NSAID, pain and sleep medication for 48 h prior to any visit, so that all measurements were taken after a 48 h medical washout period. Further exclusion criteria were MRI-incompatibility (metal in the body, claustrophobia), pregnancy, difficulties understanding Swedish, rheumatoid arthritis, metabolic disease, neurologic disease or severe psychiatric condition, malignancy, cardiovascular disease, unregulated thyroid disease, or lung disease. Participants in the HC group reported having no current pain. The clinical examination ensured that the patients met the criteria for FM according to the 1990 criteria from the American College of Rheumatology [1]. Participants in the HC group underwent the exact same procedure as the FM group, with the added inclusion criterium of reporting no current pain.

### 2.3. Materials

#### 2.3.1. Pain Assessments

Three different pain dimensions were assessed: (a) pain intensity; (b) perceived pain sensitivity and (c) pressure pain thresholds.

Pain intensity: Current pain intensity was assessed using a numeric rating scale from 0 (= no pain) to 10 (= worst possible pain). All healthy controls reported no pain. As this was a prerequisite for study inclusion, the pain intensity variable was not suitable for between-group comparison measures.

Pain sensitivity questionnaire: The PSQ consists of 17 items describing daily life situations that the participant rates on how intense pain (not aversiveness or distress) would be for them in each situation [24]. Fourteen out of the 17 items are normally considered painful (e.g., cold, sharp, blunt pain), the other three (e.g., taking a warm shower) are not, the score can range from 0 (not painful at all) to 10 (worst pain imaginable), and the mean of the 14 painful items was calculated. The PSQ was translated into Swedish using an iterative forward-backward process [25]. Although the Swedish version of the PSQ has not yet been psychometrically evaluated [25], translations to Norwegian [26], German [27], English [28], and Korean [29] have previously shown satisfactory statistical properties.

Pressure pain thresholds: Pressure pain thresholds (PPT) were determined with the use of a manual pressure algometer (Somedic SenseLab AB, Sösdala, Sweden) that was mounted with a probe (with a contact area of 1 cm^2^) on the muscle belly. The trapezius muscles bilaterally, erector spine bilaterally and tibialis anterior bilaterally were investigated. The pressure was increased by 30 kPa/s until the subject perceived pain and pushed a stop-button, or until the maximum threshold of 600 kPa. The PPT was defined as the mean of two trials obtained with minimum interval of 30 s. For details see [30,31]. The mean PPT of all sites and all measurements was calculated. 

#### 2.3.2. Psychological Distress and Disability Assessments

The following instruments to assess psychological distress and disability were answered: (a) The two subscales from the Hospital Anxiety and Depression Scale i.e., depressive (HADS-D) and anxiety (HADS-A) symptoms; (b) The Pain Catastrophizing Scale (PCS); and (c) the Pain Disability Index (PDI).

The Hospital Anxiety and Depression Scale: Both HADS-depression and HADS-anxiety subscales have seven items, the scoring range being between 0 and 21, in which a lower score indicates a lower possibility of anxiety or depression. HADS is frequently used in clinical practice and research and has good psychometric characteristics [32,33]. HADS had the clinical cut-offs: 0–6, no symptoms; 7–10, probably symptomatic; and ≥11, severely symptomatic [32]. 

Pain catastrophizing: The PCS measures three dimensions of catastrophizing: rumination, magnification, and helplessness [34,35] based on 13 items (with five alternatives). In the present study, the total PCS (PCS-total) together with the three subscales was used. The maximum score for PCS total is 52. For the subscale rumination, 16 was the total maximum score; corresponding values for magnification and helplessness were 12 and 24, respectively. For all subscales as well as the total score, a high score denotes a worse outcome.

Pain disability index (PDI): The pain disability index (PDI) is a seven-item self-report instrument based on a 10-point scale that assesses perception of the specific impact of pain on disability that may preclude normal or desired performance of a wide range of functions, such as family and social activities, sex, work, life support (sleeping, breathing, and eating), and daily living activities. Hence, PDI measures the impact that pain has on the ability of a person to participate in essential life activities on a scale from 0 to 70 [36,37].

#### 2.3.3. fMRI Assessment 

Rs-fMRI was obtained at the third visit on a Siemens Prisma 3T scanner. During the MR session, a 10-min resting state block was obtained after a task fMRI session on pleasant tactile stimulation; this task is described elsewhere [38]. The participants were instructed to keep their eyes open looking at a fixation cross that was presented on a screen, which participants viewed via goggles (VisuaStim Digital–Resonance Technologies, Northridge, CA, USA). Looking at a fixation cross requires minimal cognitive effort, which does not interfere with resting state network activity, yet it ensures that the participant does not fall asleep. For transmitting and receiving radio frequency signals, the participants were fitted with a 64 channel SENSE head coil and a single-shot echo planar imaging (EPI) gradient echo sequence was applied. The MR parameters for the functional images were: repetition time = 1030 ms, echo time = 30 ms, flip angle = 63 degrees, field-of-view = 448 × 448, voxel size = 3 mm^3^ (3 mm gap), and number of slices = 48. Structural anatomical data in the sagittal plane was obtained with a high-resolution 3D T1 weighted Turbo Field Echo scan with the following parameters: repetition time = 7 ms, echo time = 3.2 ms, flip angle = 8 degrees, field-of-view = 256 × 170, voxel size = 1 mm^3^ (no gap), and number of slices = 170.

### 2.4. Statistics

Questionnaire and PPT data were analysed using SPSS v.23.0 (IBM Corp., Armonk, NY, USA). Between-group differences were tested with two-tailed Student’s *t*-tests with a significance threshold of *p* < 0.01. Functional imaging data were preprocessed and analysed using a functional region-based seed-target and seed-voxel connectivity approach with the Functional Connectivity Toolbox (CONN) (https://web.conn-toolbox.org/) version 15 g [39] for SPM12-functions (Wellcome Department of Cognitive Neurology. London, UK) running in MATLAB (Mathworks, Inc., Natick, MA, USA). Preprocessing was done following the default pipeline in CONN and consisted of rs-fMRI data realignment; co-registration to anatomical data from each individual; segmentation of grey matter, white matter, and cerebrospinal fluid; normalization to a standard brain template in the Montreal Neurological Institute (MNI) coordinate space; and smoothing by an 8-mm full-width at half maximum Gaussian kernel. Confounding effects defined as motion regressors (calculated during realignment) were regressed out from the individual datasets, likewise scrubbing parameters to filter out excessive motion spikes (calculated with the ART-plugin in CONN with conservative settings aiming to remove 5% of normative data), white matter, cerebrospinal fluid, and a term for linear detrending; in addition, a band-pass filter (0.008–0.09 Hz) was applied to the data, as well as the component-based noise correction method (aCompCor) that is inbuilt in CONN to improve sensitivity and selectivity. 

To determine seed-target and seed-voxel connectivity values, we used the six different regions of interest (ROIs) defined in the aims as seeds (Figure 1). These ROIs were: the posterior cingulate cortex and vmPFC from the DMN, the left and right IPS from the CEN, and the left and right anterior insula from the salience network. The seed regions were extracted as functionally defined clusters from functional neural network template maps of DMN, left executive control network corresponding to left CEN [40], right executive control network corresponding to right CEN, and anterior salience network, as based on the study of Shirer et al. [41] and obtained online [42] (see Table 1).

In the CONN toolbox, connectivity measures at subject-level can be calculated with a bivariate correlation, in which the correlation was computed of the average BOLD time-series of the seed ROIs with the target ROIs (same as seed ROIs in our study) for seed-target analysis, or of the seed ROIs with each voxel of the brain for seed-voxel analysis. 

For the group-level seed-target analyses, the between-group comparison was done by entering the subject-level connectivity measures into an ANCOVA with dummy coded variables for FM and HC, and the covariates of interest (HADS and PPT) as regressors. HADS includes both the depression and anxiety scales; PPT was transformed to reflect a positive relation similar to HADS (increase in score reflects impairment) by subtracting the individual score from the overall mean. To test the effects of covariates, a multivariate regression was modelled. For this, the covariates of interest (HADS, PPT) were split into separate covariates per group, and these were centralised around the shared mean. The regression coefficients were Fisher-transformed into z-scores, which represent effect size change per unit of the measured covariate. The resulting z-scores for each group were then compared by means of F-tests and *t*-tests. For tests within the FM group, a regression analysis was performed that included HADS, PPT and pain intensity. A significance threshold was applied with an initial cluster-defining voxel-based threshold of *p* < 0.001 uncorrected and a subsequent cluster-extent threshold of *p* < 0.05. This threshold was corrected for multiple measurements by applying the false discovery rate (FDR) [43] to correct for the six target ROIs and by applying permutation tests using non-parametric statistics to correct for the six seed ROIs.

## 3. Results

### 3.1. Questionnaires and Pain Thresholds 

FM had lower PPT and higher PSQ than HC (Table 2). The FM group also scored significantly higher (*p* < 0.01) than HC on HADS-anxiety, HADS-depression, PCS total, and PDI. For the PCS subscales, the FM scored higher on helplessness, but not rumination or magnification. FM did not differ from HC on age. 

To measure psychological distress and pain sensitivity, HADS total and PPT were as the most proximate measures chosen to be used in subsequent analyses. Age did not significantly correlate with HADS total or PPT (*r* < |0.2| for either whole group or FM only).

### 3.2. Seed-Target Tests of between and within-Group Effects

Seed-target analyses were used for hypothesis testing in which connectivity between the six pre-defined ROIs was investigated. First, an ANCOVA was performed to investigate between-group differences while controlling for impact of HADS and PPT. No between-group differences were observed. As these factors showed multicollinearity, multiple regression was then performed to investigate the impact of one factor as a regressor while controlling for the other and tested with an F-test corrected for multiple measurements. No significant between-group results for these multiple regression analyses were observed.

Second, within-group effects of HADS, PPT and the FM-only variable pain intensity were investigated for the FM group using a multiple regression. The regression model showed that a higher level of connectivity strength between the right IPS and the right anterior insula (F(3,27) = 4.86, *p* = 0.0394) was associated with these variables. Post-hoc tests showed that only higher pain intensity significantly contributed to this increased connectivity strength (T(27) = 3.15, *p* = 0.0201). A subthreshold indication for a higher level of connectivity strength of right IPS with left anterior insula was observed (*p* = 0.0531). No significant effects for other seed nodes were found.

### 3.3. Exploration of Functional Connectivity ROI to Whole Brain

A seed-to-voxel analysis was used to investigate ROI connectivity to all voxels in the brain. First, between-group differences without adding covariates as regressors was tested with *t*-tests for each ROI. The right IPS showed a higher level of connectivity strength with a cluster extending from left insula to the superior frontal part of the temporal lobe for FM compared to HC (T(57) = 3.47, cluster size = 286 voxels, cluster-extent corrected *p* = 0.0131, *p* < 0.05) (Figure 2). No other ROIs showed connectivity patterns that differed between groups.

Secondly, an ANCOVA was applied to test for all seeds and covariates within the FM group only. The ANCOVA showed there was a significant interaction effect of stronger connectivity between right IPS and right precentral gyrus associated with the HADS, PPT and pain intensity covariates (Figure 3) (F(18,162) = 2.55, cluster size = 273, *p* < 0.001). This stronger connectivity showed to be driven by the covariates pain intensity (Figure 3A) (cluster size = 114, *p* < 0.001) and HADS-total (3B) (cluster size = 78, *p* < 0.001). Further post-hoc investigation of the depression and anxiety components of HADS-total showed that higher anxiety scores but not higher depression scores interacted with higher level of connectivity strength between IPS and precentral gyrus (cluster size = 47, *p* < 0.001).

## 4. Discussion

This study demonstrates changes in connectivity of the right IPS (a posterior node of the CEN) with the anterior insula and medial precentral gyrus in FM. We found that right IPS had a higher level of connectivity strength with the right anterior insula in FM patients who reported high pain intensity compared to healthy controls. Furthermore, we found a higher level of connectivity strength to the medial precentral gyrus for FM patients who reported more anxiety symptoms. Our intention with this study was to validate previous findings from small studies on connectivity of the DMN, CEN, and insula [16,17,18,19] in a larger sample size with a region-based approach. Our results confirm previous findings of a correlation between CEN-insular connectivity and higher pain intensity in FM [16]. Our study shows that specifically the right IPS node of the CEN exhibits a higher level of insular connectivity strength with the insula in relation to pain intensity. We also observed a subthreshold indication that the right IPS may have a higher level of connectivity strength with the left anterior insula as well. In a between-group seed-voxel analysis without included covariates, right IPS indeed showed stronger connectivity with left insula in FM. Thus, stronger connectivity between right insula and right IPS shows evidence of modulation by pain intensity, and an indication of a similar effect for the left insula was observed.

An abnormally functioning CEN may impair cognitive control, which can affect short-term memory and attention control; this been proposed as a possible mechanism for memory loss and attention problems in FM [44,45]. Indeed, the CEN shows signs of abnormal activation and connectivity related to FM [16,45]. Our novel finding of the positive association of anxiety with connectivity between right IPS of the CEN and the sensorimotor cortex in patients with FM may fit in with the theory of Glass and colleagues on overlapping systems for executive processes and pain processing [46]. As these authors showed, response inhibition in FM was associated with decreased activation in executive control regions, which may be associated with abnormal CEN connectivity [46]. The primary motor cortex has shown changed responses to pain in FM, such as higher levels of functional connectivity strength with the left anterior insula that was associated with increased pain [47]. However, more large-scale FM studies are necessary to confirm these results and explore the overlapping executive and pain processing systems in FM hypothesis of Glass et al. [46].

Our results do not confirm previous findings of altered DMN-insular connectivity [16,17,19], and we cannot conclude from our findings that altered DMN-insula connectivity is characteristic for FM in general, for FM who report more pain, or for FM with more psychological distress. It is noteworthy that even when further exploring our data and when lowering significance thresholds, no indication for altered DMN-insular connectivity was observed. In the present study, PPT did not correlate to any connectivity changes. A limiting factor for our study was that the psychological and pain measurements due to practical constraints were obtained at an earlier test occasion than the rs-fMRI collection. Nonetheless, our study still observed pain intensity modulation of IPS-insular connectivity that was consistent with previous findings as described above, suggesting that this may be a robust finding. A tentative suggestion may be that pain intensity is measured more sensitively to the change in IPS-insula connectivity in FM than pain sensitivity; further studies are needed to investigate this suggestion [16]. Another limiting factor was the limitation to investigating HADS, PPT, and pain intensity as only covariates. Other factors of interest that could not be investigated due to power constraints were the effects of age, other questionnaire scores, or cross-over effects from task fMRI. Age was not included as a covariate as it did not differ significantly between groups, nor did it correlate with our covariates of interest. However, future studies may combine the findings of our current study and the task-fMRI analysis reported in [38] to investigate how morphological changes, age and psychological distress and pain measurements interact in relation to functional connectivity. Another constraining factor for our study was that connectivity testing was limited to a subset of network nodes. This was, however, necessary as the dataset was noisy (due to aforementioned reasons, and heterogeneity of group in terms of age, menstrual cycle, medication, etc.) and thus only strong effects could be expected to reach significance. We therefore chose a region-based analysis of regions emerging from previous literature, to increase power by reducing the number of tested voxels without making too specific assumptions that would not adequately be supported by literature. It is clear that our approach can only address some hypotheses, and many more large-group resting state studies on FM are needed to reproduce existing findings and to address still outstanding questions. One of these questions is why the results of this investigation do not show any between-group effects on functional connectivity while controlling for HADS and PPT. As it is known that that psychological distress and pain aspects may account for functional connectivity changes, effects of psychological distress and pain in patients with FM need to be disentangled in future investigations if functional connectivity changes are to be attributed to the diagnosis of FM in specific.

Previous literature is consistently pointing to changes in the insula as one of the major leads for understanding altered neural functioning in FM [10,48,49], and findings of higher levels of connectivity strength of insula with DMN were more often reported than higher levels of connectivity strength between insula and CEN. Yet, these findings are not reproduced in our larger-scale study. The salience network, including the anterior insula, is considered to play a key role in switching from interoceptive to executive functioning, and thus, switching from activity in the DMN to activity in the CEN [50]. Menon proposed a triple network theory of the balance between the DMN, the CEN and the salience network; a disturbed balance such as an aberrantly functioning salience network may account for clinical symptoms such as pain and depressive symptoms through affecting the DMN or CEN [51]. Continuous nociceptive input that is processed by the insula has been hypothesized to play a part in damaged functioning of the salience network, which in turn may imbalance DMN and, according to the triple network theory, also the CEN [51,52]. An altered balance between networks rather than alterations in specific resting state networks may be an explanation for why the DMN did not but the CEN did show changed functional connectivity with the insula in the current study. In our previous study on chronic widespread pain (which included a few patients with FM), we found a higher level of connectivity strength of DMN with the right insula [53]. Therefore, given the current low reproduction rates of findings in chronic pain connectivity literature [48], further research may need to probe whether altered DMN-insular connectivity is a defining characteristic of fibromyalgia.

## 5. Conclusions

Compared to HC, FM showed higher levels of depression and anxiety as well as higher pain catastrophizing, pain disability, and pain sensitivity and intensity levels. We investigated functional connectivity of nodes of the CEN and DMN and bilateral insula, and present novel findings of: (1) a higher level of right IPS-insular connectivity strength associated with higher pain intensity in FM; and (2) a higher level of right IPS-precentral gyrus connectivity strength associated with higher levels of anxiety in FM. These findings strengthen the hypothesis of an imbalance between neural networks and an insular dysfunction in FM and warrant further studies investigating how such alterations relate to FM symptomatology. 

## Figures and Tables

**Figure 1 jcm-09-03560-f001:**
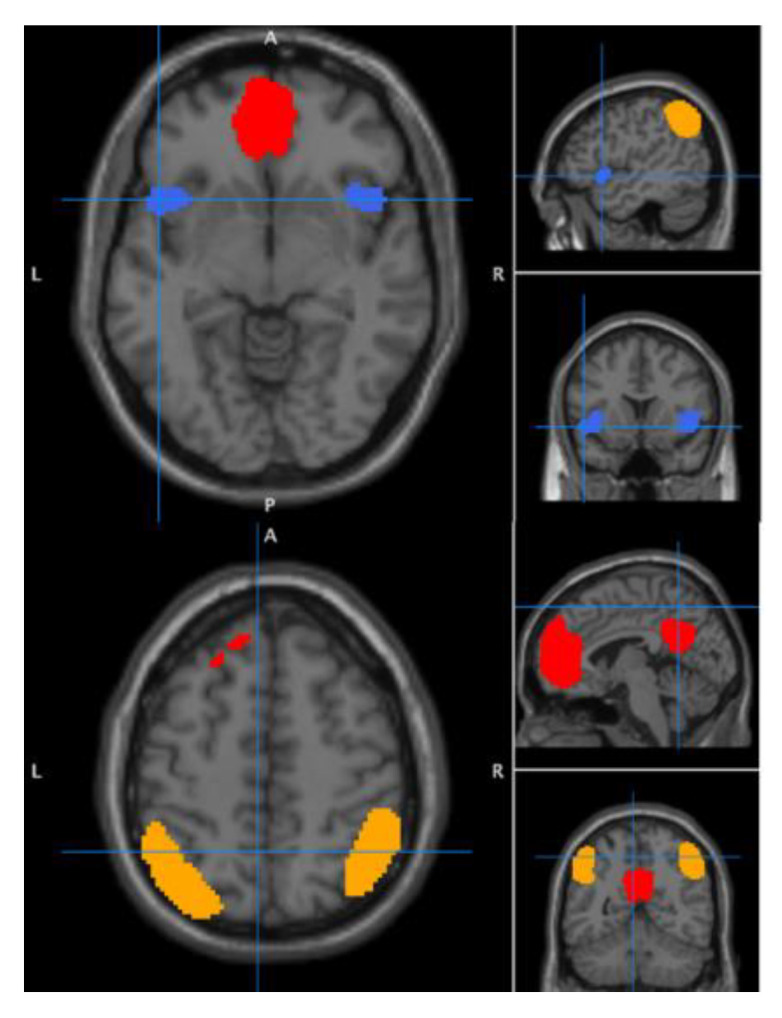
Regions of interest (ROIs) used as seeds. In red the posterior cingulate cortex and ventromedial cortex from the default mode network (A = anterior, L = left, R = right, P = posterior). In orange, the left and right intraparietal sulcus from the central executive network. In blue, the left and right anterior insula from the salience network. These ROIs were taken from published resting state networks [41].

**Figure 2 jcm-09-03560-f002:**
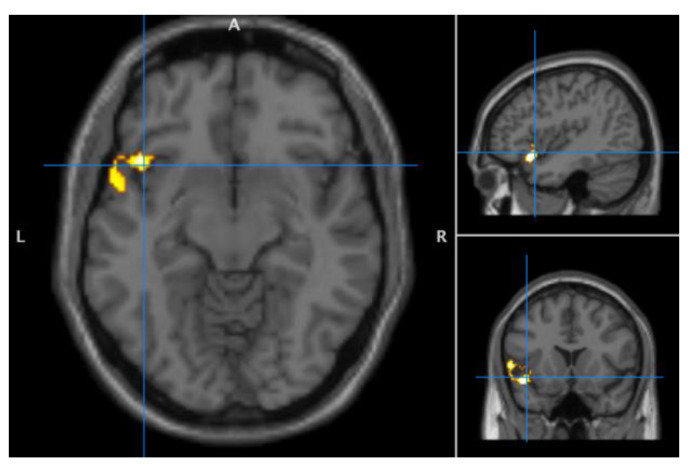
Higher level of functional connectivity strength of right posterior intraparietal sulcus (IPS) seed with left insula and temporal lobe for FM compared to HC. Seed-voxel analysis with right IPS as seed in an exploratory between-group comparison without including anxiety and depression scores, or pain sensitivity.

**Figure 3 jcm-09-03560-f003:**
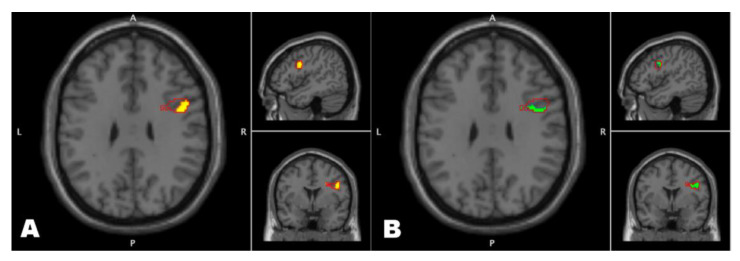
Higher levels of functional connectivity strength of right posterior intraparietal sulcus (IPS) with right precentral gyrus within the FM group were observed to interact positively with pain intensity and HADS scores. This cluster in right precentral gyrus outlined in red is the result of a seed-voxel analysis testing for anxiety and depression HADS score, pain sensitivity and pain intensity with right posterior IPS as seed. The cluster outlined in red was used as an inclusive mask for post-hoc analyses. The left side of the figure, denoted with (**A**), shows in yellow the cluster resulting from a significant positive interaction of pain intensity scores with connectivity of right IPS. The right side of the figure, denoted with (**B**), similarly shows in green the cluster resulting from a significant positive interaction of HADS scores with connectivity of right IPS.

**Table 1 jcm-09-03560-t001:** Coordinates and volume for the included regions of interest.

Region of Interest	MNI Coordinates			Volume (Voxels)
	x	y	z	
Posterior Cingulate Cortex	0	−52	30	1555
vmPFC	−4	50	14	5257
Left IPS	−42	−62	46	2110
Right IPS	46	−52	48	1873
Left anterior insula	−42	16	−2	305
Right anterior insula	42	16	0	319

MNI = Montreal Neurological Institute, vmPFC = ventromedial prefrontal cortex, IPS = intraparietal sulcus.

**Table 2 jcm-09-03560-t002:** Group statistics for questionnaires and PPT including *t*-test and corresponding *p*-values on between-group (FM vs. HC) differences.

Variable	HC (*n* = 28)			FM (*n* = 31)			Statistics	
	Mean	SD	Range	Mean	SD	Range	t	*p*-Value
Age	42.64	10.24	23–55	39.23	11.44	22–56	1.21	0.233
Pain Intensity	0	0	0	5.68	1.83	3–10	N/A	N/A
PSQ total	3.49	0.91	2.14–5.71	5.70	1.80	1.93–11.50	5.86	**<0.001**
PPT (kPa/s)	363.69	111.49	186–607	122.94	77.42	32–350	9.54 ^†^	**<0.001**
HADS total	4.14	3.14	0–12	13.81	6.75	1–29	6.93	**<0.001**
HADS-depression	1.43	1.71	0–6	6.00	3.61	0–15	6.32 ^†^	**<0.001**
HADS-anxiety	2.71	2.28	0–7	7.81	4.04	0–14	6.04 ^†^	**<0.001**
PCS total	11.32	8.98	0–45	20.32	10.47	1–45	3.53	**<0.001**
PCS rumination	5.64	3.78	0–15	6.42	3.84	0–15	0.78	0.438
PCS magnification	2.04	2.29	0–10	3.35	2.30	0–8	2.21	0.031
PCS helplessness	3.71	4.05	0–20	10.55	5.32	0–22	5.51	**<0.001**
PDI total	8.50	5.11	6–33	36.48	11.27	14–66	12.48 ^†^	**<0.001**

*n* = number of participants, SD = standard deviation, FM = fibromyalgia, HC = healthy controls, HADS = hospital anxiety and depression scale, PCS = pain catastrophizing scale, PSQ = pain sensitivity questionnaire, PDI = pain disability index, PPT = pressure pain threshold. Significant *p*-values under the threshold of *p* < 0.01 are printed in bold. ^†^ equal variances not assumed according to Levene’s test for Equality of Variances.

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
