# Peer review of "Exploration of Functional Connectivity Changes Previously Reported in Fibromyalgia and Their Relation to Psychological Distress and Pain Measures"

_jcm, 2020, doi:10.3390/jcm9113560_

Round 1

Reviewer 1 Report

The present papers aimed to investigate functional connectivity changes associated with nodes in DMN, ECN and insula, using rs-fMRI, in a group of FM patients compared to HC. Results showed a higher level of right IPS-insular connectivity strength associated with higher pain intensity in FM; and a higher level of right IPS-precentral gyrus connectivity strength associated with higher levels of anxiety in FM, whereas no change in insular connectivity with DMN was observed.

The paper has the strength of including a relatively large group of FM patients.

I have got the following suggestions:

  • Abstract: it has been reported that “The right IPS node of the ECN showed a higher level of connectivity strength with bilateral insula in FM with higher pain intensity compared to controls”. This is not totally correct since there was only “A subthreshold indication for a higher level of connectivity strength of right IPS with left anterior insula…”
  • Pag 2 lines 69-71: this sentence is not clear. I would suggest rephrasing it.
  • The “Experimental Section” is quite long. To improve the readability, I would suggest dividing it into sub-paragraphs (e.g. subjects, procedures, materials, statistics).
  • Pag 3 line 99: with “No experimental treatment was given during this large-scale study” did you mean that patients continue their standard therapy? Or that none of them changed therapy in the time between the three visits?
  • Pag 3 lines 104-105: please provide some more details regarding the HC. Did they follow the same procedure, including all the three visits?
  • Table 1: “Significant p-values under the threshold of p<0.01 are printed in bold”: there are not values printed in bold
  • Pag 6 line 234: One “)” is missing.
  • Pages 6-7, lines 231-239. It is not clear what kind of analysis has been conducted. In the text it seems that a regression has been made only on the FM Group, but then figure 2 is cited, in which reference is made to a difference between FM and HC. Furthermore, what do you mean with “A significant effect of higher level of connectivity strength was found… for these included regressors”? Which variables were included, and which variables/effects were significant?
  • Pag 7 lines 237-238: “A subthreshold indication for a higher level of connectivity strength of right IPS with left anterior insula was observed”: please provide p value.
  • Pag 7 lines 250-251: “Error! Reference source not found. 2”. Probably it should be Figure 2.
  • Pag 7 lines 255: Please replace “(F18,162) = 2.55” with ““(F(18,162) = 2.55”
  • Pag 9 lines 335-337: I understand that this is only a tentative suggestion, but I do not understand what pushed towards this hypothesis. In fact, although no alteration in DMN-insular connectivity emerged in the present work, previous studies found it in patients with fibromyalgia, not only in patients with chronic widespread pain.
  • Regarding the results, how did you explain the absence of between-group differences while controlling for impact of HADS and PPT? Between-group differences emerged only without adding covariates as regressors. Could this suggest that pain and anxiety per se, and not fibromyalgia, affect functional connectivity?

Author Response

Response to Reviewer 1

        The authors would like to thank Reviewer 1 for their kind comments and suggestions which have helped to improve our manuscript and clarify experimental details and results. We also are grateful for the thorough reading which helped spotting omissions and errors. We have responded pointwise below to the reviewer’s comments (which are in bold):

First, we would like to address that we changed terminology from “executive control network” to “central executive network”. A recent investigation from the first author showed the severe inconsistency in functional network naming of executive functions, and proposed a nomenclature based on data-driven image meta-analysis of the available body of functional network images related to executive functioning (https://www.biorxiv.org/content/10.1101/2020.07.14.201202v1.full). In hope for better reproducibility and clarity, we have adopted this nomenclature in the current paper and included a reference to this meta-analysis

  1. Abstract: it has been reported that “The right IPS node of the ECN showed a higher level of connectivity strength with bilateral insula in FM with higher pain intensity compared to controls”. This is not totally correct since there was only “A subthreshold indication for a higher level of connectivity strength of right IPS with left anterior insula…”

Response: Indeed, the phrasing should be “... a higher level of connectivity strength with right insula ...”, as those are the significant results. We apologize for the incorrect phrasing, stemming from a preliminary analysis. We changed the phrasing (line 23) accordingly.

  1. Pag 2 lines 69-71: this sentence is not clear. I would suggest rephrasing it.

Response: The sentence (now lines 73-76) has been rephrased and reads now: Higher pain intensity in three different patient groups with either chronic low back pain (n=18), complex regional pain syndrome (n=19) or knee osteoarthritis (n=14) was associated with a decreased connectivity of the vmPFC with the posterior components of the DMN 20. These patients also showed an increased connectivity of vmPFC with the insula 20.”

  1. Reviewer 1: The “Experimental Section” is quite long. To improve the readability, I would suggest dividing it into sub-paragraphs (e.g. subjects, procedures, materials, statistics).

Response: We added sub-paragraphs to “2. Experimental section” and sub-sub-paragraphs for Materials (paragraph 2.3), namely “Pain Assessments”, “Psychological Distress and Disability Assessments”, and “fMRI Assessment”.

  1. Reviewer 1: Pag 3 line 99: with “No experimental treatment was given during this large-scale study” did you mean that patients continue their standard therapy? Or that none of them changed therapy in the time between the three visits?

Response: We added the sentence “the patients continued their standard therapy” (now line 132) to clarify this issue.

  1. Reviewer 1: Pag 3 lines 104-105: please provide some more details regarding the HC. Did they follow the same procedure, including all the three visits?

Response: The healthy controls followed the same procedure, including all three visits. We clarified this as follows: Paragraph 2.2 (Procedures) at page 3 starts out with a description applicable to both patients and controls, this is now clarified by rephrasing to “all participants” (lines 122, 122). Also, we added the following sentence at the end of the Procedures paragraph: “Participants in the HC group underwent the exact same procedure as the FM group, with the added inclusion criterium of reporting no current pain.” (lines 140-141)

  1. Reviewer 1: Table 1: “Significant p-values under the threshold of p<0.01 are printed in bold”: there are not values printed in bold

Response: We apologize for losing this font style during submission preparation. We changed the values with p<0.01 to bold in now Table 2, page 7.

  1. Reviewer 1: Pag 6 line 234: One “)” is missing.

Response:  We added the previously omitted “)” at now line 297

  1. Reviewer 1: Pages 6-7, lines 231-239. It is not clear what kind of analysis has been conducted. In the text it seems that a regression has been made only on the FM Group, but then figure 2 is cited, in which reference is made to a difference between FM and HC. Furthermore, what do you mean with “A significant effect of higher level of connectivity strength was found… for these included regressors”? Which variables were included, and which variables/effects were significant?

Response: First, we would like to apologize for the confusion, the citation of figure 2 was erroneously. This paragraph is strictly about results associated within the FM group only. We removed the reference to figure 2. This multiple regression analysis tested whether any of the three variables HADS, PPT and pain intensity contributed to a model of changed connectivity. The increased connectivity strength between right IPS and right AI was significantly explained by this model including those three regressors. Post-hoc testing showed it was pain intensity that explained the model significantly while the other two variables did not contribute to the model as regressors. We changed the wording of this paragraph (lines 293-298) to explain this model in a more standardized phrasing: The regression model showed that a higher level of connectivity strength between the right IPS and the right anterior insula (F(3,27) = 4.86, p = 0.0394) was associated with these variables. Post-hoc tests showed that only higher pain intensity significantly contributed to this increased connectivity strength (T(27) = 3.15, p = 0.0201).”.

  1. Reviewer 1: Pag 7 lines 237-238: “A subthreshold indication for a higher level of connectivity strength of right IPS with left anterior insula was observed”: please provide p value.

Response: We added the corresponding p-value, line 300

  1. Reviewer 1: Pag 7 lines 250-251: “Error! Reference source not found. 2”. Probably it should be Figure 2.

Response: Indeed, this was a cross-reference to Figure 2 that broke, we corrected this error (line 311)

  1. Reviewer 1: Pag 7 lines 255: Please replace “(F18,162) = 2.55” with ““(F(18,162) = 2.55”

Response: Previously omitted bracket added (line 316)

  1. Reviewer 1: Pag 9 lines 335-337: I understand that this is only a tentative suggestion, but I do not understand what pushed towards this hypothesis. In fact, although no alteration in DMN-insular connectivity emerged in the present work, previous studies found it in patients with fibromyalgia, not only in patients with chronic widespread pain.

Response: We understand that the wording of this tentative suggestion may meet some reservations. Our intention was to reflect on the fact that if DMN-insular connectivity is indeed indicative of fibromyalgia, it is quite surprising that does not show up at even a subthreshold significance level in our current study. We rephrased this intention to a sentence that we hope to be more agreeable; “further research may need to probe whether altered DMN-insular connectivity is a defining characteristic of fibromyalgia.” (lines 428-429)

  1. Reviewer 1: Regarding the results, how did you explain the absence of between-group differences while controlling for impact of HADS and PPT? Between-group differences emerged only without adding covariates as regressors. Could this suggest that pain and anxiety per se, and not fibromyalgia, affect functional connectivity?

Response: Indeed, we find the absence of between-group differences very intriguing, and the reason to control for both HADS and PPT is because functional connectivity is known to be influenced by those factors (lines 71-75 for depression and anxiety, lines 59-70 for pain). The research on functional connectivity in relation to HADS and PPT scores was at the moment of writing to our best knowledge not extensive enough to draw a conclusion on their association with functional connectivity in the nodes we investigated. We now clarified our thoughts on this in two additional sentences: One of these questions is why the results of this investigation do not show any between-group effects on functional connectivity while controlling for HADS and PPT. It is known that psychological distress and pain aspects may account for functional connectivity changes [4,9,18,23], however, these effects in patients with FM need to be disentangled in future investigations before functional connectivity changes can be attributed to the diagnosis of FM in specific.” Lines 405-410.

Reviewer 2 Report

The manuscript by van Ettinger-Veenstra et al. describes the resting state fMRI investigation of functional connectivity changes in three neural networks (DMN, ECN, and salience) and their relation to pain and psychological distress (depression and anxiety) in fibromyalgia (FM) patients compared with the age-matched healthy controls (HC).  The authors found:

  1. Significant differences in anxiety and depressive symptoms, pain catastrophizing and pain disability index between FM patients and the age-matched HC (p < 0.0)
  2. No significant between-group difference in ROI-to-ROI functional connectivity while controlling for anxiety and depression, and pain sensitivity (PPT)
  3. Stronger functional connectivity between the right IPS ROI with left insula and temporal lobe in FM compared to HC in an exploratory ROI-to-voxel between-group comparison without including anxiety and depression scores, or pain sensitivity (Figure 2)
  4. Stronger functional connectivity between right IPS ROI and right precentral gyrus in FM compared to HC that positively interacted with pain intensity and HADS score in whole-brain ROI-to-voxel analysis (Figure 3)
  5. Correlation of anxiety symptoms in FM with higher connectivity strength between the vmPFC (DMN node) and right sensorimotor cortex
  6. No change in insular connectivity with DMN.

The paper is well written and the differences in functional connectivity and their relation to pain and HADS scores that emerged from this study are discussed well in the context of extant literature. Understanding the relation between functional connectivity and psychological distress and pain in fibromyalgia and other chronic pain conditions is an important goal, and the authors make progress toward this end. However, I have some concerns regarding the strength of the evidence presented as well as the authors' methods and interpretations, given the following issues:

1) The resting state scan was acquired after the task activation fMRI (i.e., page 4, lines 151-152: a 10-minute resting state block was obtained after a task fMRI session on pleasant tactile stimulation). Considering the growing literature that points to the effects of the prior task on the resting state functional connectivity, the rest condition might not have been a "true" rest in this case depending on how strong the prior task effects were on the networks' connectivity. In fact, the strong activation of the insula in the prior fMRI task (see reference paper 38 by the same authors), and the lingering effects of the task on DMN (a task-negative network) could have adulterated the rs-fMRI connectivity results. Can the author revisit the analysis and, e.g., break the 10-minutes resting state scan to two 5 minutes blocks and compare the results of the first 5 minutes with the last 5 minutes of the rsfMRI scan to see if the results hold?

2) The study subjects in reference paper 38 appear to comprise of the same HC and FM patients. The VBM analysis, which the authors previously performed on these subjects, showed reduced grey matter density in FM patients especially "in the bilateral hippocampus and anterior insula. This difference was even stronger when including age as a covariate of no interest" [38]. However, in the current manuscript, the authors did not consider these important findings. Possible age-related changes or individual differences in gray matter volume or density of the seed regions were not examined so it is not clear whether partial volume effects influenced the results. In fact, age was not even included as a covariate of interest with pain and HADS scores in any of the CONN ROI-to-ROI or ROI-to-voxel analyses

3) As authors acknowledged in the discussion (page 9, lines 304-306), there seems to be a relatively long time period between the first visit (clinical examination including PPT) and the third visit (MRI scan) for some of the subjects if the average time between the first and third visits was approximately 4 months +/-3 month (page 3, lines 96-98). Considering pain variability in fibromyalgia patients with time, did the authors try to look at the effect of time between the first and third visits on the outcomes especially in FM patients? Also, it is not clear exactly when pain measures and other questionnaires were taken. On page 3, line 94, "After the first visit, the participants answered…", so when exactly did the subjects take the questionnaires if not during the first visit (at the time of clinical examination)? On page 3, lines 110-113, "Pain intensity: Current pain intensity was assessed using a numeric rating scale from 0 (= no pain) to 10 (= worst possible pain)." But the time of measurement was not explicitly stated. Was the pain intensity taken at the time of the MRI scan (third visit) or during the medical examination (first visit)? In the previous paper by the same authors (reference paper 38), they also reported the average pain intensity for the previous four weeks (denoted by 'Pain Intensity 4w'). So did the authors keep track of the subjects' pain intensity e.g. by weekly questionnaires? The 'Pain Intensity 4w' could be useful in accessing the fluctuations in pain intensity.

Minor concerns:

Page 3, lines 93-96. The study design consisted of three visits: first visit for clinical examination and third visit for MRI scan. Could the authors explain what was the aim of the second visit and include the complete timeline of the study?

Page 4, lines 160 - 161. It is useful to include the parameters of the T1w anatomical scan and sequence name (MPRAGE, etc).

Page 4, lines 179 - 188. Please list all seed ROI coordinates (preferably in MNI space) and sizes (e.g., radius) that were taken from the FIND lab resting state network templates.

Page 6, lines 221-22. "Significant p-values under the threshold of p<0.01 are printed in bold". But this is not shown in the actual table.

Page 7, line 251. Please correct the broken reference.

Page 1, line 29. Considering that the "novel findings" of this study involves the executive control network (ECN) specifically the intraparietal sulcus (IPS) node of the ECN (page 9, lines 341-344), a couple of keywords that convey this conclusion is recommended.

Author Response

Response to Reviewer 2

The authors would like to thank Reviewer 2 for their time and their positive reception of our article. We are grateful for the helpful comments and suggestions. We have responded pointwise below to the reviewer’s comments (which are in bold):

First, we would like to address that we changed terminology from “executive control network” to “central executive network”. A recent investigation from the first author showed the severe inconsistency in functional network naming of executive functions, and proposed a nomenclature based on data-driven image meta-analysis of the available body of functional network images related to executive functioning (https://www.biorxiv.org/content/10.1101/2020.07.14.201202v1.full). In  hope for better reproducibility and clarity, we have adopted this nomenclature in the current paper and included a reference to this meta-analysis

  1. The resting state scan was acquired after the task activation fMRI (i.e., page 4, lines 151-152: a 10-minute resting state block was obtained after a task fMRI session on pleasant tactile stimulation). Considering the growing literature that points to the effects of the prior task on the resting state functional connectivity, the rest condition might not have been a "true" rest in this case depending on how strong the prior task effects were on the networks' connectivity. In fact, the strong activation of the insula in the prior fMRI task (see reference paper 38 by the same authors), and the lingering effects of the task on DMN (a task-negative network) could have adulterated the rs-fMRI connectivity results. Can the author revisit the analysis and, e.g., break the 10-minutes resting state scan to two 5 minutes blocks and compare the results of the first 5 minutes with the last 5 minutes of the rsfMRI scan to see if the results hold?

Response: We feel that Reviewer 2 raises a very valid point. Indeed, prior tasks may have an effect on functional connectivity, these effects may last for 5 to 15 minutes, depending on the type of effect (Tung et al., 2013 https://www.sciencedirect.com/science/article/abs/pii/S1053811913003339?via%3Dihub )

We also found that we have not a suitable way of investigating this concern. Short-term effects did presumably not affect the 10 min rs-fMRI session as there was a 5 minute break between task-fMRI and resting state: The researcher providing the tactile stimulation during task fMRI had to leave the room, then the staff running the scanner would shortly talk to the participant to see if they are ready to continue and to remind them about the upcoming sequence (to keep their eyes open and focused on the white cross.. Long-term effects in the 15 min range are likely to affect the whole rs-fMRI session. Our rationale to investigate the full 10 minutes is the clearly improved reliability of longer session lengths (>9mins) compared to shorter (5-7 min) investigations. We believe therefore that, although an arguably interesting analysis, that the power of such a 5min by 5min comparison would not be sufficient to draw any conclusions about present DMN connectivity changes in our participant group that would be reportable in this study with acceptable confidence. We do however agree with Reviewer 2 that this is a valid point of concern, that we would hope to address in future investigations. We therefore have included the following part in the discussion: “Another limiting factor was the limitation to investigating HADS, PPT and pain intensity as only covariates. Other interesting factors that were not investigated due to power constraints are the effects of age, other questionnaire scores, or cross-over effects from task fMRI. Age was not included as a covariate as it did not differ significantly between groups, nor did it correlate with our covariates of interest. However, future studies may combine the findings of our current study and the task-fMRI analysis reported in [38] to investigate how morphological changes, age and psychological distress and pain measurements interact in relation to functional connectivity.”  (lines 391-397).

We feel it is important to consider with patients with potential hyperalgesia or allodynia that whereas tactile stimulation received during the task session may have had lingering effects, this tactile stimulation was gentle and could have occurred in everyday situations as well. The process of putting a participant in an MRI scanner always involves tactile interaction, which means that any rs-fMRI, even if run before a task fMRI, would be affected by the lingering effects of tactile stimulation. Running a “clean” resting state is not very feasible unless incorporating long waiting times which may be unpleasant for patients in other ways (i.e. even the waiting and lying the scanner environment would presumably affect a following resting state).

  1. The study subjects in reference paper 38 appear to comprise of the same HC and FM patients. The VBM analysis, which the authors previously performed on these subjects, showed reduced grey matter density in FM patients especially "in the bilateral hippocampus and anterior insula. This difference was even stronger when including age as a covariate of no interest" [38]. However, in the current manuscript, the authors did not consider these important findings. Possible age-related changes or individual differences in gray matter volume or density of the seed regions were not examined so it is not clear whether partial volume effects influenced the results. In fact, age was not even included as a covariate of interest with pain and HADS scores in any of the CONN ROI-to-ROI or ROI-to-voxel analyses

Response: We agree that potential age-related effects in relation to the investigated regions are indeed a point of valid concern. It should be noted that in our investigation there was no significant age difference between groups (p=0.233). In preliminary analyses, we did investigate whether including age as a covariate would explain some of our model. It did not but including another covariate does reduce power from our planned analysis of including the significantly differing covariates of HADS, PPT and pain intensity. We now added to the Result section that age did not correlate with our covariates of interest: Age did not significantly correlate with HADS total or PPT (r<|0.2| for either whole group or FM only)” (lines 272-273).

As the reviewer likely has noticed, we could not include all measurements (e.g. PSQ, PCS, PDI) because this study was not powered to include so many covariates. We feel that, because there was no significant age difference, there was no rationale to include this as a covariate. We agree that it is a limiting factor of our study and should be investigated in the future, and therefore we included a sentence on this in the discussion (which was also addressed in the response to comment 1): “Another limiting factor was the limitation to investigating HADS, PPT and pain intensity as only covariates. Other interesting factors that were not investigated due to power constraints are the effects of age, other questionnaire scores, or cross-over effects from task fMRI. Age was not included as a covariate as it did not differ significantly between groups, nor did it correlate with our covariates of interest. However, future studies may combine the findings of our current study and the task-fMRI analysis reported in [38] to investigate how morphological changes, age and psychological distress and pain measurements interact in relation to functional connectivity.”  (lines 391-397).

It may be informative for the reviewer to know that, because of longer submission processes, the analyses in this current article were performed prior to the analyses in the cited article [38], and therefore the results from [38] could not be informative to the analyses of the current article.

  1. As authors acknowledged in the discussion (page 9, lines 304-306), there seems to be a relatively long time period between the first visit (clinical examination including PPT) and the third visit (MRI scan) for some of the subjects if the average time between the first and third visits was approximately 4 months +/-3 month (page 3, lines 96-98). Considering pain variability in fibromyalgia patients with time, did the authors try to look at the effect of time between the first and third visits on the outcomes especially in FM patients? Also, it is not clear exactly when pain measures and other questionnaires were taken. On page 3, line 94, "After the first visit, the participants answered…", so when exactly did the subjects take the questionnaires if not during the first visit (at the time of clinical examination)? On page 3, lines 110-113, "Pain intensity: Current pain intensity was assessed using a numeric rating scale from 0 (= no pain) to 10 (= worst possible pain)." But the time of measurement was not explicitly stated. Was the pain intensity taken at the time of the MRI scan (third visit) or during the medical examination (first visit)? In the previous paper by the same authors (reference paper 38), they also reported the average pain intensity for the previous four weeks (denoted by 'Pain Intensity 4w'). So did the authors keep track of the subjects' pain intensity e.g. by weekly questionnaires? The 'Pain Intensity 4w' could be useful in accessing the fluctuations in pain intensity.

Response: We apologize for being unclear on when these measurements were taken, and changed the description as follows: After the first visit, all participants answered a set of health questionnaires covering demographic data as well as pain intensity, spreading and duration and psychological characteristics at home” (line 124-125). The pain intensity 4w used in study [38] denoted the 4 weeks prior to filling in this report, there were no weekly questionnaires. We agree that this would have been very interesting to better understand our patient group, however the four visits including multiple examinations and scans were deemed taxing enough for the patients which meant that no further visits or measurements could be included.

As an effect of not having an exact date of when questionnaire and pain reports were filled in by the participant at home, we were unable to perform a reliable investigation of the effect of time between visits.

Minor concerns:

  1. Page 3, lines 93-96. The study design consisted of three visits: first visit for clinical examination and third visit for MRI scan. Could the authors explain what was the aim of the second visit and include the complete timeline of the study?

Response: We added a description of the whole study (in total 4 visits) in paragraph 2.2 at page 3. Regarding visit 2: “During a second visit blood and saliva samples were taken and a microdialysis was performed, this is not analysed in the current study.”

  1. Page 4, lines 160 - 161. It is useful to include the parameters of the T1w anatomical scan and sequence name (MPRAGE, etc).

Response: The T1w parameters and description are added to now line 202-205: Structural anatomical data in the sagittal plane was obtained with a high-resolution 3D T1 weighted Turbo Field Echo scan with the following parameters: repetition time = 7 ms, echo time = 3.2 ms, flip angle = 8 degrees, field-of-view = 256 x 170, voxel size = 1mm3 (no gap), number of slices = 170.”

  1. Page 4, lines 179 - 188. Please list all seed ROI coordinates (preferably in MNI space) and sizes (e.g., radius) that were taken from the FIND lab resting state network templates.

Response: a new table is added (now Table 1) at page 5 with the coordinates and volume of the ROIs included.

  1. Page 6, lines 221-22. "Significant p-values under the threshold of p<0.01 are printed in bold". But this is not shown in the actual table.

Response: We apologize for the loss of font style in the submission processes, the p-values <0.01 are now in bold in now Table 2, page 7

  1. Page 7, line 251. Please correct the broken reference.

Response: The correct reference to figure 2 was added at now line 311

  1. Page 1, line 29. Considering that the "novel findings" of this study involves the executive control network (ECN) specifically the intraparietal sulcus (IPS) node of the ECN (page 9, lines 341-344), a couple of keywords that convey this conclusion is recommended.

Response: Many thanks for this helpful suggestion. We added the keywords “central executive network” and “intraparietal sulcus”
